# Anonymous Traffic Detection Based on Feature Engineering and Reinforcement Learning

**DOI:** 10.3390/s24072295

**Published:** 2024-04-04

**Authors:** Dazhou Liu, Younghee Park

**Affiliations:** Faculty of Computer Engineering, Charles W. Davidson College of Engineering, San Jose State University, San Jose, CA 95192, USA; dazhou.liu@sjsu.edu

**Keywords:** Tor, anonymous traffic, feature engineering, unsupervised learning, reinforcement learning

## Abstract

Anonymous networks, which aim primarily to protect user identities, have gained prominence as tools for enhancing network security and anonymity. Nonetheless, these networks have become a platform for adversarial affairs and sources of suspicious attack traffic. To defend against unpredictable adversaries on the Internet, detecting anonymous network traffic has emerged as a necessity. Many supervised approaches to identify anonymous traffic have harnessed machine learning strategies. However, many require access to engineered datasets and complex architectures to extract the desired information. Due to the resistance of anonymous network traffic to traffic analysis and the scarcity of publicly available datasets, those approaches may need to improve their training efficiency and achieve a higher performance when it comes to anonymous traffic detection. This study utilizes feature engineering techniques to extract pattern information and rank the feature importance of the static traces of anonymous traffic. To leverage these pattern attributes effectively, we developed a reinforcement learning framework that encompasses four key components: states, actions, rewards, and state transitions. A lightweight system is devised to classify anonymous and non-anonymous network traffic. Subsequently, two fine-tuned thresholds are proposed to substitute the traditional labels in a binary classification system. The system will identify anonymous network traffic without reliance on labeled data. The experimental results underscore that the system can identify anonymous traffic with an accuracy rate exceeding 80% (when based on pattern information).

## 1. Introduction

Anonymous networks play an important role in safeguarding user identities and privacy. Existing anonymous networks include onion routing (Tor) [1], garlic routing [2], the Mix Network [3], and the Invisible Internet Project (I2P) [4]. Notably, Tor and the Mix Network are two common types of anonymous networks. Tor routes data through a series of intermediary nodes, in which it emphasizes preventing the associations of communication partners [1]. On the other hand, the Mix Network employs multiple routers to shuffle and randomize encrypted messages [3]. By using these networks, both ends can communicate without sharing identities such as IP addresses.

The demand for using anonymous networks has increased. For instance, approximately two million users accessed Tor in the first quarter of 2020 [1]. In legitimate cases, users may access services like file sharing while preserving anonymity. On the other hand, anonymity can result in misbehavior in cyberspaces, which includes cyber attacks. For example, packets targeting the Darknet are suspicious and usually generated by malware or attackers searching for vulnerabilities [5]. Moreover, investigations into the Tor network unveiled that malware and counterfeits have been circulating [6]. The authors proposed to detect distributed denial of services (DDoSs) by evaluating Darknet traffic [7]. Not only do attackers exploit anonymous networks to attack non-anonymous entities, but the Tor network itself is subject to denial of service (DoS) attacks, which may lead to the loss of bandwidth resources [8]. Furthermore, attempts to deanonymize clients, locate servers within the Tor network, and reduce the service availability of the Tor network have become the targets of attackers [9]. These trends imply that attackers have abused anonymous networks, although there are legitimate use cases. Therefore, efforts to detect anonymous traffic are necessary for preventing adversarial affairs, and our work assumes anonymous traffic is an anomaly.

Blocking the IP addresses that are involved in anonymous traffic has been one tactic [10]. When encountering suspected anonymous traffic, the IP address of the incoming traffic is compared against each blocked IP address. However, this approach can be expensive and performs poorly whenever the list is outdated [11]. Alternatively, in research, there have been attempts to trace clients that use the Tor network. In this study, the researchers controlled a Tor router and a server in the Tor network to generate purposefully modified packets. When the controlled Tor router observes modified packets, the client’s identity can be linked and confirmed. Nonetheless, this approach requires control over Tor routers and over the deployment of servers in the Tor network, which can undermine the Tor network’s functionalities. In summary, these aspects demonstrate that detecting anonymous traffic remains a challenge.

To improve the efficiency of traffic analysis, researchers have inspected anonymous traffic by focusing on packet characteristics and stream patterns [12]. Recently, machine learning and deep learning-based methods have been explored for analyzing anonymous traffic. Extensive studies have been performed on supervised learning [12,13,14]. In supervised learning approaches, the objective is to learn an approximation of P(y|x), where the given datasets are represented as ${\left[x_{i},y_{i}\right]}_{i=1}^{n}$, where $x_{i}$ is the feature set and where $y_{i}$ denotes the corresponding label [15]. Classifiers, a subset of supervised learning algorithms, are heavily employed to compute boundaries that categorize samples into distinct regions. In the deep learning field, classifiers are constructed with complex structures that are capable of processing features in high dimensions. While classifiers achieve high performance on static data, those types of models exhibit shortcomings upon deployment in real time. For instance, their real-time deployment in the realm of IDS shows vulnerability to the obfuscated features of dynamic network traffic [16]. The performance evaluation of some machine learning- and deep learning-based models displayed a high false positive rate in classifying Darknet traffic [13]. Moreover, due to the limited availability of public datasets, researchers have needed help accessing data about anonymous network traffic [17,18]. Consequently, training detection models can be challenging.

Regarding the existing challenges, the motivations can be summarized as follows:The generation of anonymous traffic, often linked to adversarial activities on the Internet, underscores the necessity of its detection for maintaining cybersecurity.While supervised learning algorithms are widely utilized to examine anonymous traffic, especially classifiers designed for predictive tasks, they still face challenges. These challenges include limitations in adapting to unpredictable network conditions and significant time and resource demands, such as a lengthy training time and dependence on labeled data. By applying feature engineering, we focus on extracting packet- and timing-related features, which are crucial for differentiating between anonymous and non-anonymous traffic.By addressing the adaptability of the Markov decision process (MDP) to dynamic environments, we implement a classification system based on the MDP principles and outcomes of feature engineering. The result is a robust classification system that can efficiently detect anonymous traffic with a simplified architecture.

Based on the motivations, the contributions are summarized as follows. In this work, we differentiated between anonymous and benign network traffic by incorporating packet and timing features. These features distinguish effectively between anonymous and benign traffic, as well as pave the way for subsequent experiments in anonymous traffic detection. Furthermore, we applied the Markov decision process (MDP) framework to the classification system, thereby enhancing decision-making tasks and system resilience by strategically designing actions, states, and rewards. We also introduced a classification model characterized by a simplified structure with reduced resource demands. The efficiency of these reductions is validated through decreased training times and performance evaluations. An accuracy rate exceeding 80% is anticipated by leveraging our framework’s architecture.

We arrange the structure of the paper as follows. Initially, Section 2 delves into the Tor network and Darknet technical details. This section also introduces supervised learning-based detection approaches and explains the applications of reinforcement learning frameworks in the network domains. Subsequently, Section 3 discusses the proposed reinforcement learning-based system. Following Section 3, Section 4 explains the system development and testing processes. Finally, the paper is concluded by summarizing critical findings related to the proposed system.

## 2. Related Work

### 2.1. Overview of the Tor Protocol and Darknet Protocol

#### 2.1.1. The Tor Network

The Tor network is a distributed overlay network aiming to anonymize TCP-related activities such as web browsing and message sending [1]. Further, the Tor network consists of relay nodes run by volunteers spanning the globe [19]. Each relay node operates as a router that facilitates the reception of incoming traffic and the routing of outbound traffic to the intended destination.

The Tor network routes through a chain of no less than three relay nodes. Each relay node in the chain is executed as a regular process and knows only its immediate predecessor and successor [1]. This design ensures that no intermediary can deduce the identity of the destination entity from the source entity and vice versa.

The Tor network encrypts and decrypts data layer by layer. Specifically, plain texts are encrypted to create a ciphertext. The ciphertext is encrypted once more. Similarly, the ciphertext is decrypted layer by layer until the plain text is derived. When the client, e.g., the Tor browser, transmits data through the Tor network, the local proxy encrypts the data layer by layer. When a relay node in the Tor network receives data, it decrypts one of the layers.

Due to the routing and encrypting mechanism, data routed through the Tor network attains high anonymity and privacy. In summary, the Tor network has been designed to refrain from traffic analysis and tracing. When attackers abuse the Tor network, the target entities risk being attacked.

#### 2.1.2. The Darknet Protocol

The surface web is a subset of the web that indexes web content on the publicly available part of the Internet, which can be accessed using standard search engines such as Bing [20]. In addition to visiting surface webs, the Tor network contains servers hosting web pages, which are referred to as the Onion Services. The servers’ network supporting Onion Services can be considered the Darknet. As the Darknet is restricted to be accessed through tools such as the Tor network, the Tor browser has become one gateway to the Darknet.

#### 2.1.3. Features of Darknet and Tor Protocol

Differentiating Darknet and Tor traffic facilitates the detection of malevolent anonymous traffic. By examining the open source codes of the Tor browser [19], we identified four differences between the Darknet protocol and the Tor protocol. These differences include the locations of destination servers, the number of relay nodes in a circuit, IP addresses, and the DNS resolution time.

For the location of destination servers, the Tor protocol directs user data to servers outside the Tor network. In contrast, the Darknet protocol routes user data to servers within the Tor network. The feature of the circuit length describes the number of hops to reach a public server or a Darknet server. The path to a public server consists of three Tor relay nodes, while the path to a Darknet server contains six. As for the IP address, the Darknet servers conceal their IP address from the clients. Visiting servers outside the Tor network requires a DNS resolution through the local proxy or at the exit relay node on behalf of the client. However, in general, Darknet protocols do not involve DNS resolutions.

Table 1 summarizes the four features.

Based on investigations of the Darknet protocol, ten features were identified as Darknet traffic characteristics. The ten features are listed in Table 2:The first feature is related to irregular domain names. Owing to the Darknet protocol’s encryption features, the domain name of a Darknet website is typically derived from an encryption key (e.g., AES) and represented as a random string.The second feature is about relay node bandwidth. Due to the distributed architecture of the Tor network, relay nodes in the Tor network are expected to route a vast amount of network traffic. As a result, the bandwidth of each Tor router can be significantly higher than the bandwidth of a regular home router.The third feature is the size of the packets transmitted within the Darknet. Notably, these packets, or cells, have a fixed length of 512 bytes.Regarding server locations, Darknet servers are typically housed within the Tor network, meaning that developers often deploy them on relay nodes within the Tor network. The identification of servers in unusual placements can serve as an indicator of Darknet traffic. Such server location anomalies help distinguish Darknet activities from regular Internet traffic.Darknet servers are not expected to interact with public users on the publicly available part of the Internet.Attackers tend to use irregular port numbers to deceive intercept policies.A circuit is a path formed by relay nodes from the client to the server. As for the circuit length, the circuit length originating from the client to a Darknet server is typically six, while the circuit length from the client to an external server is three [21].The eighth feature is related to the Domain Name Service (DNS) and is summarized as the DNS delay time. Conventionally, the domain names or Uniform Resource Locators (URLs) are resolved by the DNS to IP addresses, and known DNS queries can be cached to improve efficiency and scalability [22]. Since visiting websites through different relay nodes in the Tor network may require new resolutions, Darknet and Tor traffic are expected to exhibit a long DNS resolution time.The IP addresses of the Darknet tend to be unreachable since these IP addresses are typically not assigned to legitimate servers [23]. Consequently, attackers can engage in malicious activities such as backscatter for vulnerabilities by utilizing these unassigned IP addresses.Data flows dealing with Tor network servers often contain more packets compared to those involving external servers. This discrepancy could be due to attackers sending more packets within the Tor network to obscure their malicious activities.

According to the ten features outlined in Table 2, the Darknet protocol attained high anonymity. Hence, forensic analysis of Darknet traffic is inherently complex. A more viable approach is to focus on pattern analysis. Nonetheless, the values of these ten features are unavailable due to the limitations of capturing real-time Darknet traffic with tools such as the Wireshark packet analyzer. In future work, we aim to extract pattern information to distinguish between malicious anonymous and benign traffic.

### 2.2. Detecting Tor Traffic by Supervised Learning

As discussed in Section 2.1, reading or tracing anonymous Tor traffic is highly complicated. This aspect poses challenges to identifying Tor traffic by packet inspections. To address this issue, researchers calculated flow-level features and developed supervised learning-based approaches.

Lashkari et al. generated eight types of network traffic (e.g., browsing, chat, streaming) and captured Tor traffic between the client and entry node [24]. In their approach, they extracted eight categories of timing-related statistics. Those features include the time between the arrival of two packets, the time a flow remains active or idle, flow duration, and the number of bytes or packets in one second. The Zero Rule, C4.5 decision tree, and K Nearest Neighbor were applied to classify network traffic into the Tor or benign classes. Based on the analysis, the C4.5 decision tree can detect 93.4% of the Tor samples. These findings indicate that timing-based features contribute to unveiling Tor traffic patterns, even without accessing the packet contents.

To address the reliance of many classifiers on extensive labeled training data, an enhanced decision tree algorithm was introduced [25]. The authors identified four specific features that serve as the unique characteristics of Tor traffic. These four features are entropy related to packet length, the frequency of appearance of packets with a length of 600 bytes, the number of packets with zero data in the first ten packets, and the average time between the arrival of two packets. For the supervised model, a decision tree was constructed. Instead of adopting the splitting attributes in traditional C4.5 and ID3 decision trees, information gain is employed to select the most informative attributes. Specifically, attributes with the highest information gain are chosen as the splitting attributes. The authors collected network traffic and gathered 50,000 samples during the testing phase. The results indicate that the modified decision tree achieved an accuracy of up to 99% for detecting Tor traffic.

Due to privacy concerns and the limited availability of the dataset on anonymous traffic, researchers often gather private data or generate traffic within simulated environments [17]. In response to this challenge, the authors introduced the Anon17 dataset, logging features associated with Tor and other anonymous network traffic instances [17]. These features encompass packet header information, packet counts, and the length in bytes of each flow. Leveraging the Anon17 dataset, the researchers probed how ML techniques can identify anonymous traffic [18]. The classification algorithms include Naive Bayes, Bayesian Networks, C4.5, and random forest. The results show that classifiers can classify Tor and other types of network traffic instances with an accuracy close to 100%.

Most supervised models for detecting anonymous traffic rely on classification tasks. The efficacy of these models depends on the quality of data preprocessing, feature engineering, and the training process. When trained with sufficient data and crafted features, those models can detect anonymous traffic with an accuracy of over 90%. Conversely, the performance of supervised models may degrade due to inadequate data and non-relevant features. This aspect prompts the exploration of alternative decision-making paradigms.

### 2.3. Reinforcement Learning

Reinforcement learning involves sequential decision making. In sequential decision making, the goal includes learning what actions to take at a state to maximize the expected returns. The mapping of a state to an action is referred to as a “policy”. Unlike instructive approaches, reinforcement learning-based methods involve the assessment of their behaviors. The evaluations of policies are quantified in terms of a value or the probability of taking an action at a state.

#### 2.3.1. Value-Based Learning

Upon following a policy, the policy is evaluated by value. The value metric represents the expected cumulative return of being in a state. In value-based learning, the value of a state can be updated iteratively. When the environment model is not unknown, model-free learning is an approach to solving MDP problems, for example Q-learning and State–Action–Reward–State–Action (SARSA). Q-learning and SARSA utilize Q-values to learn optimal policies that lead to the highest expected returns by updating temporal difference (TD) errors. The TD error is as follows [26]:(1)$$Q\left(S,A\right)\leftarrow Q\left(S,A\right)+\alpha [R+\gamma max_{a}Q\left(S^{\prime },a\right)-Q\left(S,A\right)].$$

The Q-value of the action taken at the current state is updated by the TD error of $R+\gamma max_{a}Q\left(S^{\prime },a\right)-Q\left(S,A\right)$. Equation (Equation 1) is practical in scenarios where the state–action space is discrete. To solve problems involving continuous state–action spaces, researchers studied substituting tables with neural networks [27]. A Deep Q Network is introduced to learn policies for controlling Atari 2600 games. Within their model, Q-values are parameterized as $Q(s,a;\theta_{i})$, where $\theta_{i}$ is the weight set of a convolutional neural network. The temporal difference formula was harnessed to update the parameters of the neural network. When evaluating the Deep Q Network using Atari games, their framework outperforms human players and baselines in 29 games in terms of game scores.

#### 2.3.2. Applications in the Network Environment

Reinforcement learning frameworks have many applications for optimizing the network topology and developing counter-attack strategies.

Researchers developed a reinforcement learning-based approach to make optimal routing decisions while satisfying security requirements [28]. In the proposed method, each state represents a switch on the data plane of the Software-Defined Network. In addition, security devices are deployed on some of the switches. The agent’s goal is to traverse from the source switch to the destination switch while avoiding paths with high latency, jitter, and packet drop rates. The proposed method utilizes a Q table and defines the reward function as the weighted sum of the delay, jitter, traffic rate, and packet loss. Compared to existing link stability-based Q-routing, the results exhibit a reduced delay time, regardless of the number of security constraints.

A Deep Q Learning technique is applied to counter jamming attacks in cognitive radio networks (CRNs) [29]. In a CRN system, the participants include primary users (PU), secondary users (SU), and jammers. The proposed method represents each state as the appearance of PUs and the signal-to-interference-plus-noise ratio (SINR). The agent is an SU, and its action is to choose to either leave the jamming area or defeat the jammer. A deep convolutional neural network is leveraged to approximate the Q-values by addressing the limitations of Q learning to ample state space. The result shows that the proposed method has a faster convergence time and achieves a higher SINR than the naive Q learning-based method.

According to the applications of reinforcement learning, we define the actions, states, and rewards so that these definitions fit the characteristics of a finite and discrete state–action space.

## 3. Methodology

This section explains the network environment in which the proposed detection system will be deployed and the adapted MDP framework to detect anonymous traffic.

### 3.1. Network Environment

#### 3.1.1. Pattern Information Processing

In network environments, detecting anonymous traffic in real time is challenging due to its encryption features. Rather than examining each packet, our system identifies patterns based on sequences of packets, or ‘flows’, characterized by consistent source and destination IP addresses and ports [12]. Within this context, the pattern information is related to packet statistics, such as flow length in bytes and timing statistics, including the flow duration and time elapsed between the arrival of consecutive packets. As real-time training data are unavailable, the system computes these features as soon as it captures a flow. For instance, to determine the average packet size within a flow, it divides the total byte counts by the total packet counts.

#### 3.1.2. Flow Pattern Visualization

Figure 1 and Figure 2 visualize the differences in the timing and packet patterns of anonymous and regular traffic flows, respectively. The horizontal axis marks the number of samples, while the vertical axis records the feature value of each sample. The feature values are standardized to unit variance to minimize the bias imposed by the maximum and minimum values.

Figure 1 displays the mean packet length in a flow. It can be seen that the feature values of approximately 40,000 samples vibrate in the range between 0 and 12.5. For regular traffic, the majority of samples range from zero to five, although some feature values exceed five along the vertical axis.

Figure 1 displays the mean packet length in a flow. In the figure, the feature values of approximately 40,000 samples vibrate between 0 and 12.5. Most samples range from zero to five for regular traffic, although some feature values exceed five along the vertical axis.

Figure 2 displays one of the timing-based features, or the mean inter-arrival time of two consecutive packets in the forward direction. The plot of anonymous traffic in Figure 2 indicates that the feature values of approximately 40,000 anonymous traffic samples range between 0 and 14. For regular traffic, the feature values of all regular traffic samples range from zero to eight.

The results imply that anonymous traffic flows exhibit different timing and packet-based patterns than non-anonymous traffic. Hence, analyzing the packet and timing-based features of anonymous traffic paves the road toward detecting anonymous traffic.

### 3.2. MDP Environment Descriptions

MDP is a standard framework for addressing sequential decision-making problems [30]. An MDP framework consists of a quadruple denoted as (S, A, r, p). In the quadruple, element S represents a state space with a finite set of states, A represents the action space with a finite set of actions, r is the reward value received upon executing an action, and p is a transition probability. Subsequently, the proposed detection system simulates the MDP framework, as depicted in Figure 3. In Figure 3, the process involves selecting a set of features, e.g., the average length of packets in a flow, by the system, according to the current policy. Next, following the execution of action $A_{t}$, the environment sends the reward to the agent, resulting in a transition to another state. This design addresses the interactions between the agent and the environment.

**State S:** The state space contains three states: Tor, non-Tor, and ambiguous. The state space is interpreted as S={s1,s2,s3}. In *S*, s1 represents a vague state, s2 represents a Tor state, and s3 represents a non-Tor state. The initial state of the agent is ambiguous, meaning that the agent has no clue about the behavior of the detected traffic. As such, the agent’s goal is to identify an answer to leave the ambiguous state. Accordingly, the Tor and non-Tor states are considered goal states. The agent assumes that the Tor network generates the current traffic when reaching the Tor state. The current traffic flow is recognized as non-Tor when transiting to the non-Tor state.This design replaces the conventional labels used for classification tasks with these model states, allowing the system to process the observed network traffic differently.**Action A:** The agent’s action is to select features. The action space is defined as $A=\{a_{1},a_{2},a_{3},\ldots ,a_{n}\}$, where $a_{i}\in A$ and $a_{i}=\left\{f_{1},f_{2},f_{3},\ldots ,f_{t}\right\}$. The subscript n represents the number of actions, and t is the number of features each action selects.To rank features based on feature importance, the feature selection techniques provided by the *Scikit-learn* library [31] are used to select features. Those selection techniques include Recursive Feature Elimination (RFE), Recursive Feature Elimination with Cross-Validation (RFECV), the random forest classifier, the mutual information-based SelectKBest library, and Support Vector Machine (SVM). The rationale for selecting these techniques will be revealed in Section 4.Based on this design, the action space size is five. The first, second, third, fourth, and fifth actions select features ranked by RFE, RFECV, the random forest classifier, the SelectKBest API with mutual information, and SVM. Table 3 summarizes the action space. We reveal and discuss the feature rankings and the selected features in Section 4:**Heuristic reward function:** The heuristic reward function reflects an agent’s action. Our system calculates the reward value by taking the linear summation of weighted feature values as the input to the hyperbolic tangent function. This design of the reward function bounds the reward values within the range between −1 and 1. Precisely, the scalar value of the immediate reward is calculated as:
(2)$$r\left(s,a\right)=\tanh\left(w_{1}\times f_{1}+w_{2}\times f_{2}+\ldots +w_{t}\times f_{t}\right)\;$$In Equation (Equation 2), parameter w represents the weight of each feature, and variable f is the feature value. The scalar sum of each product of the feature value and the corresponding weight is the input to the hyperbolic tangent function. Specifically, the formula of the hyperbolic tangent function is as follows:
(3)$$\tanh\left(x\right)=\frac{e^{x}-e^{-x}}{e^{x}+e^{-x}},$$
where *e* is the natural exponent and the variable *x* is the input to the hyperbolic tangent function, or $\left(w_{1}\times f_{1}+w_{2}\times f_{2}+w_{3}\times f_{3}+\ldots +w_{t}\times f_{t}\right)$ in Formula (Equation 2). Finally, the reward function is written as follows:
(4)$$r\left(s,a\right)=\frac{e^{\left(w_{1}\times f_{1}+\ldots +w_{t}\times f_{t}\right)}-e^{-(w_{1}\times f_{1}+\ldots +w_{t}\times f_{t})}}{e^{\left(w_{1}\times f_{1}+\ldots +w_{t}\times f_{t}\right)}+e^{-(w_{1}\times f_{1}+\ldots +w_{t}\times f_{t})}}.$$**Environment model:** The model of the environment is the transition probability. The conditional probability of transiting to the next state is P(s_next|s,a). Based on an environment model, an agent’s interactions with the environment can be model-based or model-free. For instance, the environment model in balancing a pendulum can be approximated by knowledge about kinematics. However, an explicit model, given the definitions of state, action a, and reward functions, has yet to be identified among the recorded network flows in CIC-Darknet2020 [14]. Thus, the agent is expected to make decisions and interact with the environment without being aware of an environment model.

### 3.3. Threshold Setting

A two-threshold system distinguishes between Tor and non-Tor traffic flow samples, with each threshold tuned explicitly to its respective traffic type. Setting these thresholds aims to calibrate the system’s sensitivity toward identifying Tor traffic. For example, a higher threshold for Tor traffic may decrease the sensitivity, leading to fewer Tor traffic flows being detected. Another goal is to minimize the dependence on labels. Instead of contrasting the computed results with predefined labels, the system compares the results with tuned thresholds. This methodology enables the system to decide whether a specific result exceeds or falls below the set threshold, which facilitates the identification of traffic types in a lightweight manner.

Figure 4 illustrates the horizontal axis on which the Tor and non-Tor thresholds are tuned. Since the experiment considers Tor samples as the Positive Class, the endpoint with the value of 1 denotes Tor samples, and the endpoint with the value of −1 denotes non-Tor samples. Additionally, the value of the Tor threshold is greater than that of the non-Tor threshold. Besides, the segmentation forms three intervals. The first interval is located on the left side of the non-Tor threshold. The middle interval is located between the non-Tor and Tor thresholds. The third interval is situated on the right side of the Tor threshold. Samples with reward values that are distributed on the right side of the Tor threshold are identified as Tor. On the other hand, samples distributed on the left side of the non-Tor threshold towards −1 are detected as non-Tor traffic. Lastly, the middle interval indicates ambiguous samples, meaning the system cannot extract sufficient information to decide. In the above configuration, the system distinguishes Tor and non-Tor flows in an unsupervised mode by setting the thresholds, and the labels are substituted with threshold values.

Note that the distances from each threshold to either endpoint do not contribute to the decision-making process.

### 3.4. System Diagram

Initially, the agent is placed in the ambiguous state in Figure 5 and selects the first action in the action space specified in Table 3. After choosing the first action, the environment assigns the immediate reward, and the agent compares it with the two thresholds. There are three comparison outcomes: lower than the non-Tor threshold, higher than the Tor threshold, and higher than the non-Tor threshold, but lower than the Tor threshold (the middle interval in Figure 4).

Figure 5 illustrates the transition diagram consisting of three states (Tor, Non-Tor, and Anonymous), where each vertex represents a distinct state and each edge denotes a transition triggered by an action, or (*a*), along with the associated reward value, or (r(s,a)).

### 3.5. Detection Algorithm

The MDP framework is implemented as a modified transition probability set. Based on the equation below, the probability of transiting to an ambiguous state is one if the reward is less than the Tor threshold and more significant than the non-Tor threshold. The probability of transiting to the Tor state is one if the reward value exceeds the Tor threshold. The transition probability of transiting to the non-Tor state is one if the reward value is lower than the non-Tor threshold. On the other hand, in either the Tor or non-Tor state, the agent reaches the goal and re-initiates in the ambiguous state.
$$p\left(s_{next}|s,a\right)=\left\{\begin{array}{cc} p(s_{next}=ambiguous)=1, & nonTor<r(s,a_{t})<Tor-threshold\\ p(s_{next}=Tor|s=ambiguous,a_{t})=1, & r(s,a_{t})\geq Tor-threshold\\ p(s_{next}=nonTor|s=ambiguous,a_{t})=1, & r(s,a_{t})\leq nonTor-threshold. \end{array}\right.$$

Algorithm 1 describes the transitions between states according to the actions, rewards, states, state transitions, and thresholds. In the adapted MDP framework, the agent aims to arrive at either a Tor or non-Tor state from the ambiguous state. After executing an action, a reward value higher than the Tor threshold or lower than the non-Tor threshold triggers a decision, and the agent departs the ambiguous state. After departing the ambiguous state, the workflow of the current traffic concludes. In turn, the agent moves back to the ambiguous state to analyze the upcoming traffic flows. However, subsequent actions, i.e., the second through the fifth action, are selected until the agent exits the ambiguous state. If the agent remains ambiguous after selecting all five actions, the agent cannot decide and neglects the traffic flow.

Ambiguous traffic flows are recorded in a buffer. Elements in the buffer require further analysis, depending on factors such as respective Intrusion-Detection System policies.
**Algorithm 1** Transitions between states.1:**procedure** Transition(threshold_Tor,threshold_nonTor,feature_set,batch_size)                 ▹ Function2:    **for** *i* in range(0, len(feature_set), batch_size) **do**    // Create batches3:        batch_features=feature_set[i:i+batch_size]    // Calculate reward value of each sample in the batch4:        r_batch=first_action(batch_features)5:        **for** *j* in *range*(len(batch_features), batch_size) **do**6:           **if** r[j]>=threshold_tor **then**                          ▹ Classified as Tor7:               State←Tor8:           **else if** r[j]<=threshold_non_tor **then**                  ▹ Classified as non_Tor9:               State←Non_Tor10:           **else**                                ▹ Classified as Ambiguous11:               **while** r[j]>threshold_non_torandr[j]<threshold_tor **do**12:                   **if** noactionsremaininginactionspace **then**13:                       Label(Anonymous&&Malicious)              ▹ Further processing required14:                       break15:                   State←Ambiguous16:                   r_batch=next_action(batch_features)                ▹Selectnextaction

## 4. Experiment

This section delineates the development and testing of the proposed method. The raw dataset was preprocessed in the development phase, and features were ranked according to the feature engineering techniques. In the last stage of the development phase, we trained five single-layer perceptrons and utilized each perceptron’s corresponding weights in the input layer as the reward function’s parameters in Equation (Equation 2). In the testing phase, the system’s performance was gauged based on the accuracy, recall, and precision. Lastly, the system performance was compared against the performance of conventional classifiers.

### 4.1. Dataset Processing

#### 4.1.1. Raw Dataset

The CIC-Darknet2020 dataset is a labeled dataset that summarizes the statistics of bidirectional anonymous and non-anonymous traffic flows [14]. We used these historical statistics to develop the MDP framework.

The researchers extracted packet- and timing-based features using the *CICFlowMeter* [14] tool. Timing-based features include packet inter-arrival time, idle time, and bytes per second. Packet-based features include pattern information about the packet size and contents.

There are four types of labels: non-Tor, non-VPN, Tor, and VPN. We consider VPN traffic anonymous because VPN traffic is encrypted, although VPN traffic is tunneled and generated by different means than that of the Tor traffic. Therefore, the Tor and VPN labels are merged as Tor types, while the non-Tor and non-VPN labels are incorporated into the non-Tor type, resulting in two labels.

Table 4 summarizes the raw dataset.

To elaborate, the number of flow records is 141,530, while the number of features is 83. The number of flows labeled as non-Tor and Tor is 117,219 and 24,311, respectively.

#### 4.1.2. Cleaning and Preprocessing

The raw dataset was converted to a pandas dataframe for preprocessing and cleaning.

In the cleaning phase, samples that contain infinite values and “Not a Number” entries were dropped. Then, the features that share the same value among all samples were discarded. The resulting number of features was 62, and the number of samples was reduced from 141,530 to 141,483. To further simplify the feature set, the Feature Selection with Variance Thresholding (VarianceThreshold) technique in the *Scikit-learn* library (abbreviated as *sklearn*) [31] was applied. Specifically, the VarianceThreshold technique removes features with a variance below a specified threshold. In the experiment, the threshold was set at 30%. After running the VarianceThreshold, four features were further eliminated, resulting in 58 features.

The label distributions of the raw dataset are skewed, as shown in Table 4. Basically, samples labeled as non-Tor are quite abundant, while the number of Tor samples is 24,311 and only accounts for 17.18% of the 141,483 samples. To balance the dataset, the non-Tor samples whose indexes range from 0 to 69,999 were eliminated. Next, we generated 20,000 synthetic data by duplicating the samples labeled as Tor or VPN.

After cleaning and balancing, the finalized dataset contains 91,483 samples and 58 features. As Table 5 displays, the final dataset contains 47,172 Tor samples and 44,311 non-Tor samples, both of which account for approximately 48.4% of the 91,483 samples. Lastly, the number of features is 58.

#### 4.1.3. Label Encoding

Since there are two types of labels, the Bernoulli Equation was applied. The Bernoulli Equation is defined as P(X=0)=1−p and P(X=1)=p, where *X* is a random variable and p is the probability of *X* being equal to 1. As a result, an integer of either 0 or 1 replaces each categorical label. Since anonymous traffic is defined as an anomaly, each Tor label is replaced with the integer 1 (Positive Class), and each non-Tor label is replaced with the integer 0 (Negative Class). This step was achieved through the Label Encoding method in the *sklearn* library.

However, to facilitate the separations between non-Tor and Tor samples, the labels belonging to the Negative Class were converted from 0 to −1. To perform the conversion, the formula applied is represented as follows: label_encoding=2×label_encoding−1. After applying the formula, the labels in the Positive Class are represented as the integer 1, and the labels in the Negative Class are replaced with the integer −1.

### 4.2. Action Space Construction

The dimensionality of our action space is two, encompassing a multitude of actions. Each action within this space is defined as a feature selection technique in the *sklearn* library, and each action selects a specific set of features. Upon processing the raw dataset, it is imperative to finalize the total number of actions available within this action space along with the specific number of features selected by each individual action.

#### 4.2.1. Determining the Number of Actions

According to Table 3, the random forest classifier (RF), Recursive Feature Elimination (RFE), Recursive Feature Elimination with Cross-Validation (RFECV), mutual information, and Support Vector Machine (SVM) were chosen for defining each action and utilized to rank the feature importance.

In the experiment, a higher importance ranking indicates that a feature is more likely to contribute to the predictive performance. Since the feature importance ranking is based on importance scores, the experiment adopts RF to generate the importance scores and measure the amount of information contained in each feature to reduce the uncertainty. Also, RF was applied to the RFE and RFECV APIs as their score function. Subsequently, the RFE and RFECV techniques were leveraged to recursively eliminate features containing less information about the target variables. Additionally, the mutual information technique was employed to measure the relevance between a feature and the target variable. Lastly, as 58 features are involved, SVM was utilized to reduce over-fitting and improve robustness.

The experiment has determined the composition of the action space, which now comprises five actions: Random Forest (RF), Recursive Feature Elimination (RFE), Recursive Feature Elimination with Cross-Validation (RFECV), mutual information, and Support Vector Machine (SVM).

#### 4.2.2. Determining the Number of Features Selected by Each Action

After determining the number of actions, the number of features selected by each action was determined. When selecting the number of features for each action, one explicit method is to include as much information as possible, for example feeding all of the 58 features. However, this method may incur undesired data volume. Another approach is to include fewer features. To determine the optimal number of features selected by each action, we evaluated the accuracy by using the random forest classifier. The random forest classifier was chosen as it generates importance scores. In addition, it is used as the score function of the Recursive Feature Elimination and the Recursive Feature Elimination techniques with Cross-Validation in the experiment.

The classification accuracy of the random forest classifier was measured regarding the number of features. In the evaluation process, initiating the experiment with 15 features, the number of features was increased gradually in increments of 15 until all 58 features were used. Figure 6 depicts the resulting accuracy curve. The accuracy improved from 0.9746 when selecting 15 features to 0.9774 when selecting 30 features. Nonetheless, the accuracy saturated when more than 30 features were selected. This trend indicates that the accuracy improves as the number of features is increased within a specific range. However, selecting more features did not improve the performance further. Therefore, the number of features selected by each action was determined to be 15. This decision balances between maximizing input information while minimizing complexity and computational demands.

#### 4.2.3. Determining the Specific Features Selected by Each Action

After determining the number of features each action should select, the experimentation focused on identifying which 15 out of the 58 features would be most appropriate for each action. We explored two main strategies: randomly selecting 15 features from the 58 available features and the selection based on the highest importance scores. The random selection approach revealed that the number of possible combinations for selecting 15 features from 58 is approximately $2.97\times 10^{13}$, a figure that is impractical to solve due to its vast scale. To address this challenge, we adopted a second approach that prioritizes feature selection based on importance scores. This method utilized feature selection APIs in the *sklearn* library, incorporating techniques such as Recursive Feature Elimination (RFE), Recursive Feature Elimination with Cross-Validation (RFECV), mutual information, Random Forest Classification, and Support Vector Machine (SVM). By selecting the top-15 ranked features as determined by each technique, the action space was refined to consist of five actions, each selecting 15 features with the highest importance scores. This strategy concluded the construction of the action space.

### 4.3. Feature Selection Process

The feature selection process was initiated by searching for the most appropriate parameter set. The grid search on the random forest classifier was applied. The parameters to be searched are displayed in Table 6.

The result of the grid search is as follows. The number of estimators is 125, the maximum depth 15, the minimum number of samples to split a node 2, and the minimum number of samples in a leaf node 1.

We used the resulting configuration of the random forest classifier as the score function of RFE and RFECV. The number of folds of the cross-validations in RFECV was set to three. Subsequently, the random forest classifier with the same parameter configuration, as mentioned in Table 7, was built based on the processed dataset. As for the SelectKBest API, the mutual information technique was used as its score function. Finally, an SVM with a linear kernel was built based on the processed dataset.

By applying these feature selection techniques, the 15 features with the highest 15 ranks were derived. Table 8 displays the ranking results.

For RFE, the most crucial feature was identified as *Bwd Packet Length Min*, while the feature with the 15th rank was *Idle Min*. Similarly, for RFECV, the most crucial feature was *flow duration*, while the feature ranked 15th was *Flow IAT Mean*. Furthermore, there are standard features that RFE, RFECV, SelectKBest, and RF select. Features such as *Bwd Packet Length Min*, *Flow Packets/s*, and *Flow IAT Max* were consistently determined as informative for distinguishing between Tor and non-Tor traffic.

### 4.4. Reward Function Experimentation

At first, the reward function was defined as a linear equation. Furthermore, the endeavor was to compute the weights of the linear equation such that the aggregate of weighted feature values separates the Tor and non-Tor samples around a threshold of zero. Nonetheless, the calculations imply that the correspondence between the weights and the labels was intricate. Therefore, non-linearity was introduced to facilitate the separation of Tor and non-Tor samples.

#### Hyperbolic Tangent-Based Reward Function

A single-layer neural network was harnessed to implement the reward function, as mentioned in Equation (Equation 2). Specifically, the weights of the reward function were derived from the trained neural network. The activation function at the output layer serves to add the non-linearity. In each instance of the single-layer neural network, the input layer contains 15 neurons, while the output layer has 1 neuron. The hyperbolic tangent function is employed at the output layer to map the linearly weighted summation of feature values onto a single value.

Prior to the training process, the feature values of each sample underwent standardization to reach unit variance. The reason for executing standardization is that many feature values have a vast range. For instance, the *idle max* feature has a maximum value of $1.44\times 10^{15}$, whereas the *Bwd Packet Length Min* feature has a maximum value of 1350. Accordingly, the *StandardScaler* class of *sklearn* [31] was leveraged to bring all feature values to the same scale.

During the training phase, each neural network was trained in a supervised manner using the *PyTorch* library. Stochastic gradient descent (SGD) with a learning rate of 0.001 was employed as the optimizer. It was observed that the loss stopped decreasing after 30 epochs. Consequently, each neural network was trained within 30 epochs in a trial. However, another observation indicated that the weights in each training trial varied, even when the same feature set was applied. To determine the influence of those variations on the detection accuracy, the performance of each trained neural network was tested. The result exhibited that varying weights across trials did not reduce the accuracy, provided the feature sets were used consistently across trials. As a result, the weights produced by the last run were finalized as the weights of each reward function. The results of the reward functions are recorded in Table 9.

Regarding the correlations between the rankings of feature weights and the rankings of feature importance, the results showcase that the rankings of the weights did not align with the rankings of feature importance generated by the *sklearn* feature selection techniques. For instance, within the feature set selected via RFE, *Idle Min* had the lowest ranking and the third-highest weight of 0.1763. Meanwhile, *Flow IAT Max* was ranked at the top position by SelectKBest. However, it had the third-highest weight. This discrepancy suggests that the weights in a neural network and the importance of features are different metrics in evaluating the contributions of the features to predictive performance.

Based on Table 9, the reward function corresponding to each action is represented as $R\left(s,a_{t}\right)\;=\tanh\left(W_{t}^{T}\times \;a_{t}\right)$. Hence, the reward of a1 was calculated as $R\left(s,a_{1}\right)\;=\tanh\left(W_{1}^{T}\times \;a_{1}\right)$, where the weight set was transposed and multiplied by the features selected by action a1. According to Equation (Equation 4), the reward value of selecting the features ranked by RFE is calculated as
(5)$$r\left(s,a_{1}\right)=\tanh\left(1.2904\times \mathrm{Bwd}\;\mathrm{Packet}\;\mathrm{Length}\;\mathrm{Min}+\ldots +0.1763\times \mathrm{Idle}\;\mathrm{Min}\right).$$

### 4.5. Threshold Probing

#### 4.5.1. Adjusting Threshold Values

Adjusting the sensitivity to Tor traffic requires setting appropriate thresholds. We initiated with the first action set to select features as ranked by Recursive Feature Elimination (RFE), with the expectation that the target thresholds would be close to 0. The probing began at this baseline. A nested loop was employed for the probing process, where the outer loop adjusted the non-Tor threshold from −1 to 1 and the inner loop varied the Tor threshold within the same range, both utilizing an incremental step of 0.02. The results from this threshold probing indicated that the optimal accuracy was achieved when the non-Tor threshold was set at −0.3.

As depicted in Figure 7, the highest accuracy was reached in the range between −0.25 and 0.2. Taking into account the width of this ambiguous interval and the intersection point of accuracy, recall, and precision metrics, the Tor threshold was set at −0.3. Consequently, the Tor and non-Tor thresholds were finalized at −0.1 and −0.3, respectively, marking the end of the threshold optimization process.

#### 4.5.2. Adjusting the Action Sequence

We adjusted the sequence of actions to be applied at the ambiguous state to improve accuracy. Different from Table 3, the action space can be arranged as in Table 10.

The first action (Number 1) in the sequence had the highest impact on accuracy to the extent of 5 to 10%. However, adjusting the action sequence while keeping the first action fixed trivializes the accuracy improvement.

The experiment recorded the maximum accuracy of positioning the RFE, RFECV, mutual information, random forest classifier, or SVM at the first. Figure 8 shows the trends. Arranging Recursive Feature Elimination (RFE) as the first position in the action space yielded the highest accuracy at 82%, whereas employing features ranked by the random forest classifier led to the lowest accuracy, recorded at 72.2%.

The final action space is defined as $A=\{a_{1},a_{2},a_{3},a_{4},a_{5}\}$, where a1 is RFE, a2 is RFECV, a3 is mutual information, a4 is SVM, and a5 is random forest.

### 4.6. Model Deployment and Testing

The workflow of the testing procedure is divided into four phases as shown in Figure 9.

In phase 1, the saved and trained perceptrons were loaded from the .pth files, and the training process was detailed in Section 4.4. Phase 2 involved extracting the features listed in Table 8 from the cleaned dataset. These features were then standardized for processing in the proceeding phases. In the third phase, a supervised testing procedure was conducted. Specifically, a reward value was obtained in every step, and the reward value was compared with the pre-defined thresholds. The state to which the agent transits was then compared with the known label for the current sample. If the state and label matched, the sample was correctly identified. Conversely, the samples would be misclassified if the state and label did not match. In the final phase, the model performance was gauged based on the precision, recall, and accuracy metrics. The results of these metrics are recorded in Table 11.

Additionally, we visualize in Figure 10 the reward value distributions of Tor and non-Tor traffic samples when the Tor threshold was set at −0.1 and the non-Tor threshold was set at −0.3. Figure 10 demonstrates that the Tor and non-Tor thresholds effectively separated the reward values of Tor and non-Tor samples into two regions.

### 4.7. System Performance Comparisons

In addition to model testing, the performance of the proposed system was assessed against other baselines by the accuracy. The baselines included the CNN model (DeepImage) described in [14], the stacking ensemble model developed by [13], the random forest classifier implementation in our experiment, the improved decision tree algorithm (Tor-IDS) mentioned in [25], and the random forest tested on the Anon17 dataset [18]. Table 12 lists the comparison results.

The comparison results highlight the outstanding performance of supervised baselines in identifying Tor traffic patterns. Among these models, the improved decision tree algorithm [25] and the random forest implementation [18] had the highest accuracy at up to 0.99, followed closely by the ensemble model at 0.98, the random forest classifier in our experiment at 0.97, and the DeepImage model at 0.95. It is worth noting that, while the proposed model trailed behind these supervised models in terms of accuracy, it achieved an accuracy level exceeding 80%. This discrepancy indicates that ensemble and decision tree-based models in fully supervised modes can process static data and predict patterns with a high efficiency. In contrast, the proposed model targets operating in an unsupervised mode. This strategy addresses the challenges posed by the absence of labeled data in real-time network environments. As a trade-off, the proposed model was developed with a single dataset and will function in the absence of the labeled dataset upon deployment. Furthermore, each instance of the single-layer neural network has a simple structure and rapid training time. With continued improvements, the accuracy is considered satisfactory in this study. Ultimately, those aspects make deployments of this system in real-time situations more practical and robust.

## 5. Conclusions

Anonymous networks preserve user anonymity by relaying data through a distributed network. Using tools such as the Tor network makes the transmitted data hard to read and trace for third parties. However, traffic originating from anonymous networks can be suspicious or malevolent. Therefore, the real-time detection of anonymous traffic is crucial, yet inherently complex. This work has implemented a real-time system for detecting anonymous traffic using a labeled dataset containing 141,530 samples. Instead of relying on traditional classifiers, features most informative about Tor and non-Tor traffic were extracted. These important features were then used to construct a tuple comprising information about the system’s state, action, reward, and transition. Based on this tuple, we implemented a system with a simplified structure utilizing a single-layer feed-forward neural network for classification tasks. During deployment, the system continuously monitors network flows and analyzes relevant features. It makes decisions by reward signals and compares each reward signal against predefined thresholds. Depending on the comparison results, the agent transitions to one of the following states: the Tor, Non-Tor, or ambiguity. In the testing phase, the model was gauged in a supervised manner. The result indicates that the model’s accuracy for detecting anonymous traffic is 82%, which could have implications for network security and privacy.

In future work, we aim to develop robust methods for detecting traffic generated by Darknet protocols within the Tor network, along with distinguishing between benign and malicious anonymous traffic. This endeavor would involve refining the analytical techniques, which may contribute to strengthening cybersecurity measures.

## Figures and Tables

**Figure 1 sensors-24-02295-f001:**
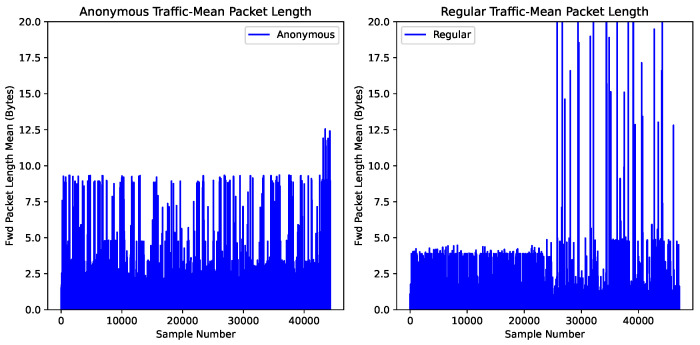
Packet dynamic of flows.

**Figure 2 sensors-24-02295-f002:**
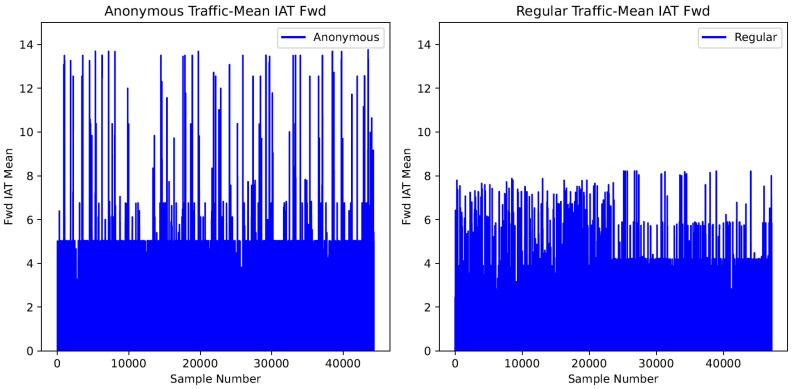
Timing dynamic of flows.

**Figure 3 sensors-24-02295-f003:**
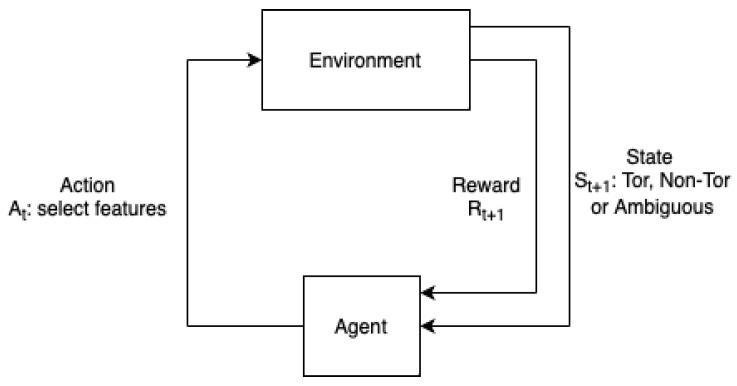
MDP scheme description.

**Figure 4 sensors-24-02295-f004:**
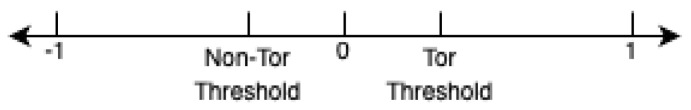
Threshold interval.

**Figure 5 sensors-24-02295-f005:**
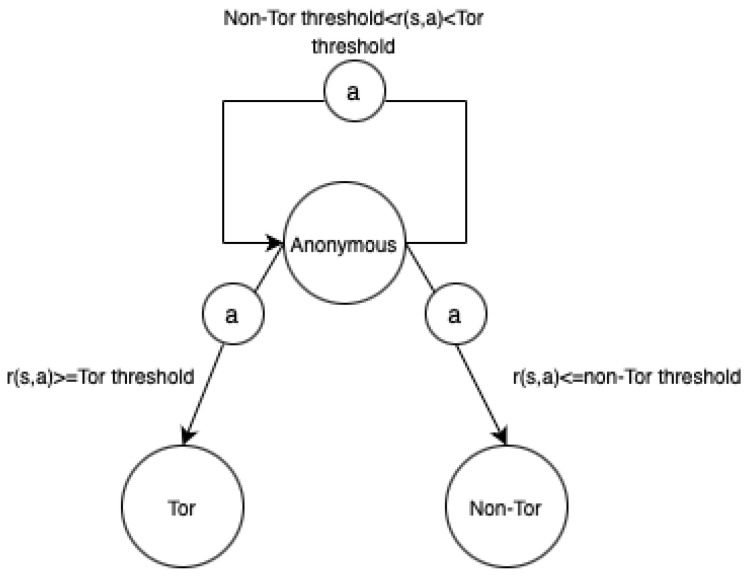
State transition diagram.

**Figure 6 sensors-24-02295-f006:**
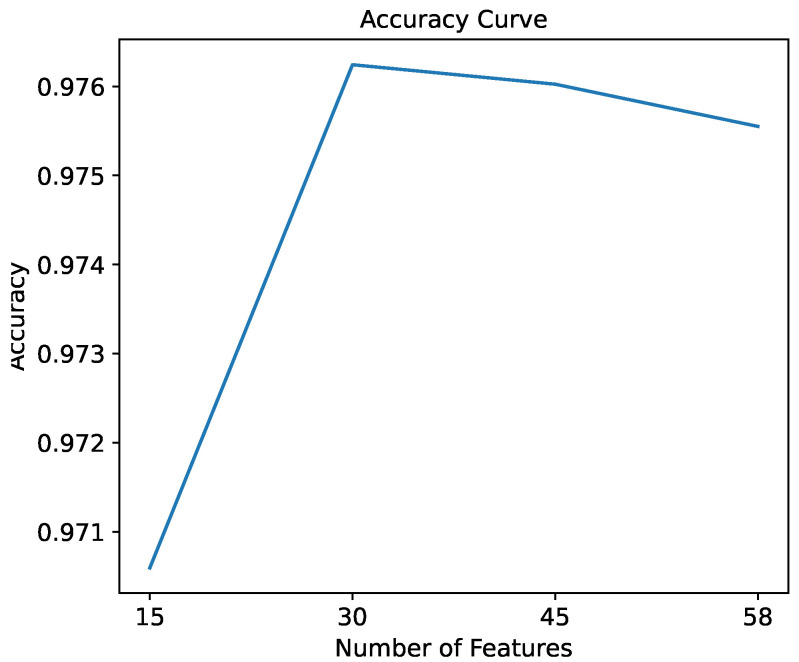
Accuracy trend of the random forest classifier regarding the number of features.

**Figure 7 sensors-24-02295-f007:**
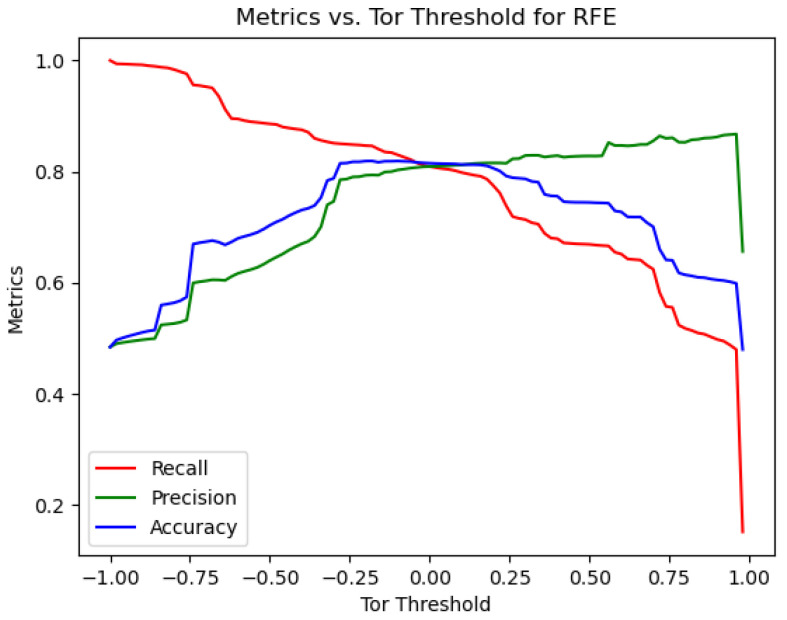
Accuracy, precision, and recall trends regarding the Tor threshold Adjustments.

**Figure 8 sensors-24-02295-f008:**
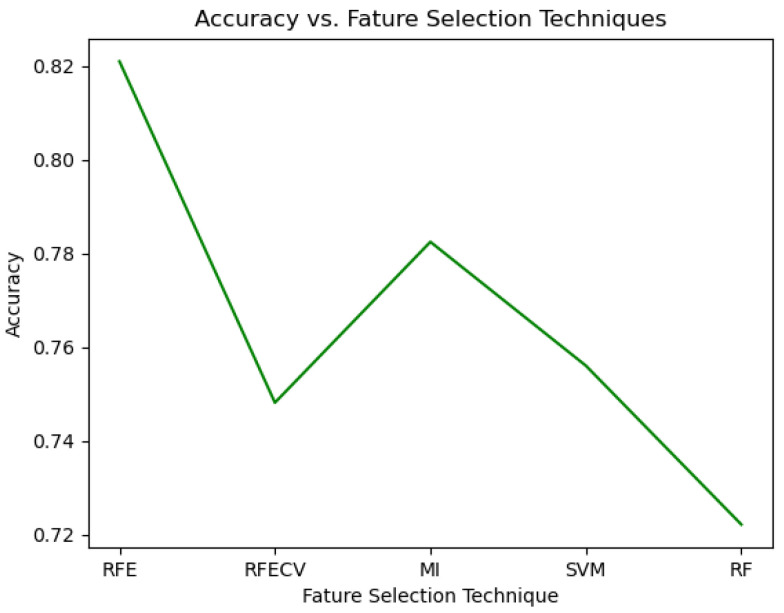
Accuracy curve of feature selection techniques.

**Figure 9 sensors-24-02295-f009:**
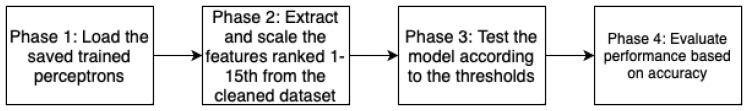
Phases of the deployment and testing procedure.

**Figure 10 sensors-24-02295-f010:**
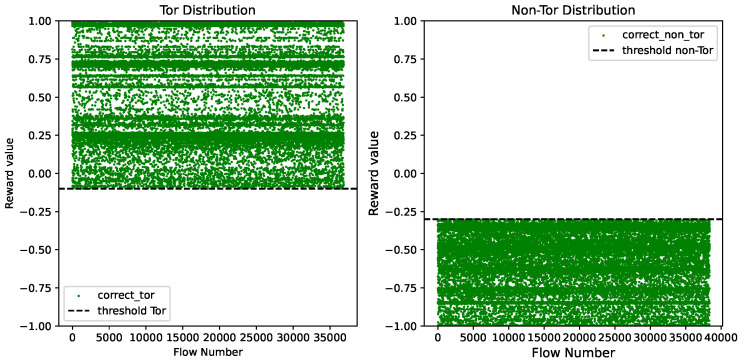
Reward distributions of Tor and non-Tor samples.

**Table 1 sensors-24-02295-t001:** Darknet protocol vs. Tor protocol.

	Server Location	Circuit Length	Server IP Address	DNS Resolution
**Tor protocol**	Outside the Tor network	3 relay nodes	Public	By clients or exit relay nodes
**Darknet Protocol**	Within the Tor network	6 relay nodes	Concealed	No resolution

**Table 2 sensors-24-02295-t002:** Ten characteristics of the Darknet traffic.

Number	Feature Name	Number	Feature Name
1	Domain name	6	Range of port numbers
2	Relay node bandwidth	7	Number of relay nodes
3	Cell (packet) size	8	DNS delay time
4	Server location	9	Invalid IP addresses
5	No interactions with external servers (Passive)	10	Quantity of cells

**Table 3 sensors-24-02295-t003:** Action space.

Number	Action
1	Select features ranked by RFE
2	Select features ranked by RFECV
3	Select features ranked by random forest classifier
4	Select features ranked by mutual information
5	Select features ranked by SVM

**Table 4 sensors-24-02295-t004:** Raw dataset.

Total Samples	Total Features	Non-Tor Samples	Tor Samples
141,530	83	117,219	24,311

**Table 5 sensors-24-02295-t005:** Raw dataset.

Total Samples	Total Features	Non-Tor Samples	Tor Samples
91,483	58	47,172	44,311

**Table 6 sensors-24-02295-t006:** Search space for random forest classifier.

Number of Estimators	Maximum Depth	Minimum Number of Samples to Split a Node	Minimum Number of Samples in a Leaf Node
75	5	2	1
100	10	5	2
125	15	10	4

**Table 7 sensors-24-02295-t007:** Results of grid search.

Number of Estimators	Maximum Depth	Minimum Number of Samples to Split a Node	Minimum Number of Samples in a Leaf Node
125	15	2	1

**Table 8 sensors-24-02295-t008:** Feature rankings by feature selection techniques.

Rank	RFE	RFECV	SelectKBest	RF	SVM
1	Bwd Packet Length Min	Flow Duration	Flow IAT Max	Flow IAT Min	Bwd Init Win Bytes
2	Bwd Packet Length Mean	Total Length of Fwd Packet	Flow Duration	Idle Max	Bwd Packet Length Std
3	Flow Packets/s	Total Length of Bwd Packet	Flow IAT Mean	Bwd Packet Length Min	Fwd Packets/s
4	Flow IAT Mean	Fwd Packet Length Min	Flow Packets/s	Flow IAT Mean	PSH Flag Count
5	Flow IAT Max	Bwd Packet Length Max	Fwd Packets/s	Bwd Segment Size Avg	Packet Length Min
6	Flow IAT Min	Bwd Packet Length Min	Bwd Packets/s	Subflow Bwd Bytes	Flow Packets/s
7	Fwd Header Length	Bwd Packet Length Mean	Flow IAT Min	Flow Packets/s	Bwd IAT Total
8	Bwd Packets/s	Flow Bytes/s	Flow Packets/s	Bwd Packet Length Mean	Down/Up Ratio
9	Bwd Segment Size Avg	Flow Packets/s	Average Packet Size	Idle Mean	Subflow Fwd Bytes
10	Subflow Bwd Bytes	Flow IAT Mean	Packet Length Mean	Fwd Header Length	Bwd Packet/Bulk Avg
11	FWD Init Win Bytes	Flow IAT Std	Packet Length Max	Flow IAT Max	Bwd Packets/s
12	Fwd Seg Size Min	Flow IAT Max	Packet Length Std	Bwd Packets/s	FWD Init Win Bytes
13	Idle Mean	Flow IAT Min	Packet Length Variance	Flow Bytes/s	Bwd IAT Min
14	Idle Max	Fwd IAT Total	Bwd Segment Size Avg	Fwd Packets/s	Fwd Packet Length Min
15	Idle Min	Fwd IAT Mean	Bwd Packet Length Mean	Idle Min	SYN Flag Count

**Table 9 sensors-24-02295-t009:** Reward function parameters.

Rank	Weight Set of a1 (RFE)	Weight Set of a2 (RFECV)	Weight Set of a3 (SelectKBest)	Weight Set of a4 (RF)	Weight Set of a5 (SVM)
1	1.2904	0.2689	0.2543	−0.0083	2.6751 × $10^{-3}$
2	−0.0694	0.1872	0.6159	−0.2216	$-3.2666\times 10^{-2}$
3	−0.0365	−0.1440	−0.3229	0.1781	$2.8813\times 10^{-1}$
4	−0.0984	0.0584	−0.0131	0.1014	$1.2825\times 10^{-2}$
5	−0.0583	−0.1191	−0.2443	−0.1772	$6.0968\times 10^{-1}$
6	−0.3272	0.7849	−0.1869	−0.1262	$1.9927\times 10^{-1}$
7	0.1684	0.1817	−0.2665	−0.0275	$2.2302\times 10^{-1}$
8	−0.2140	−0.5075	0.1950	−0.0723	$-4.0300\times 10^{-3}$
9	−0.0785	−0.0699	−0.2263	−0.0743	$-1.5365\times 10^{-1}$
10	0.1425	−0.1312	−0.4022	−0.1299	$-2.0000\times 10^{-2}$
11	0.0707	−0.2003	−0.2983	0.1927	$-2.6902\times 10^{-1}$
12	0.4345	−0.1743	−0.1267	0.0887	$3.4434\times 10^{-1}$
13	0.0277	−0.2615	−0.3179	0.2001	$-5.2559\times 10^{-4}$
14	−0.2260	−0.0326	0.4655	−0.1554	$6.3868\times 10^{-1}$
15	0.1763	0.2427	0.6879	−0.2383	$-3.8217\times 10^{-2}$

**Table 10 sensors-24-02295-t010:** Action space Alteration.

Number	Action
1	Select features ranked by SVM
2	Select features ranked by RFECV
3	Select features ranked by random forest classifier
4	Select features ranked by mutual information
5	Select features ranked by RFE

**Table 11 sensors-24-02295-t011:** Model performance evaluation.

Accuracy	Precision	Recall
0.82	0.8	0.83

**Table 12 sensors-24-02295-t012:** Model performance evaluation.

Model	Architecture	Dataset	Accuracy
Proposed Real-Time System	Single-Layer Perceptron	CIC-Darknet2020	0.82
DeepImage [14]	CNN	CIC-Darknet2020	0.95
Random Forest by Grid Search	Decision Trees	CIC-Darknet2020	0.97
Ensemble Model [13]	Decision Trees and KNN	CIC-Darknet2020	0.98
Tor-IDS [25]	Decision Trees	Self-Collected Network Traffic	0.99
Random Forest [18]	Decision Trees	Anon17	0.99

## Data Availability

The original data presented in the study are openly available at [https://www.unb.ca/cic/datasets/darknet2020.html (accessed on 1 December 2023)].

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
