# Peer review of "Anonymous Traffic Detection Based on Feature Engineering and Reinforcement Learning"

_sensors, 2024, doi:10.3390/s24072295_

Round 1

Reviewer 1 Report

Comments and Suggestions for Authors

The overall writing of the paper is good, I have a few small problems need to be revised:

1. The introduction should highlight the main contributions and innovations of the proposed method.

2. Is the reinforcement learning algorithm used in the paper applicable? Which reinforcement learning algorithm is used? Can Tor state space be represented by Q table?

3. What is the convergence of the proposed algorithm?

4. The conclusion should be cut down.

Comments on the Quality of English Language

Minor editing of English language required

Author Response

Hi, 

Thank you so much for the comments and feedbacks. The manuscript has been edited to correct grammar errors and for improved readabilities.

Regarding the reinforcement learning part, the system is still a binary classifier based on neural networks. The emphasis lies in designing a MDP framework, which are defining the states, actions, and rewards. However, we did not implement a specific algorithm, such as the Q learning or dynamic programming, to reach to an optimal policy from a random policy. This is mainly due to the difficulty in approaching a proper transition probability among samples. By using the designed MDP framework, we reduced the system architecture to a single layer neural network, while having a 82% accuracy over 91,483 samples in a semi-supervised manner. 

As for the references, the relevance of the value-based learning part in the related work section appears ambiguous to me. But the purpose is to comprehensively explain the reinforcement learning framework and foreshadow the designs of the MDP framework, although the value-based learning is not implemented.

The conclusion has been shortened to include only the proposed work and future work.

Reagrds,

Dazhou Liu

Reviewer 2 Report

Comments and Suggestions for Authors

The paper entitled "Anonymous Traffic Detection based on Feature Engineering and Reinforcement Learning," was written in good shape, whoever I have some technical comments and suggestions.

1-

  1. The methodology for detecting anonymous traffic using reinforcement learning and feature engineering is innovative. However, it could benefit from a clearer explanation of the selection criteria for features and the decision-making process within the reinforcement learning model. Specifically, elaborating on how features were prioritized and the rationale behind choosing certain actions over others in the reinforcement learning framework would enhance understanding.

2-

  1. The experimental setup and validation process are well-designed, but the paper would benefit from a more detailed comparison with existing methods. Including a broader range of benchmarks and a deeper analysis of the model's performance under varying network conditions would provide a stronger evidence base for the effectiveness of the proposed approach.

3- 

  1. The paper discusses anonymous traffic detection but does not deeply address the potential for adversarial evasion techniques. A discussion on how the system might adapt or be made resilient to such techniques would be a crucial addition, considering the adaptive nature of threats in network security.

  2.  

Author Response

Hi,

Thank you for the affirm and valuable suggestions.

  1. For selection criteria, the features with the highest rankings based on importance scores are prioritized. The rankings are generated primarily through importance scores. As a result, the 15 features with the highest importance scores are prioritized. To utilize importance scores, random forest classifier, random forest classifier-based recursive feature elimination, and recursive feature elimination with cross-validation were used. The Mutual Information is also chosen, since the amount of dependence between features is also of interest. For SVM, it is tested to separate the class to different regions. Overall, the actions are chosen over other actions such that the classification accuracy can reach the highest.
  2. Benchmarking is one critical factor to showcase the model performance. However, due to limited access to other paper's implementation details, only the declared evaluation results were used to compare against the proposed model. Since the proposed model is a classification system, accuracy and precision were primarily used for evaluation.
  3. Initially, the plan is to test and analyze the model by using real data about anonymous traffic. To achieve this, on the local laptop, we attempted to collect traffic generated by the Tor browser. However, no meaningful data is collected. The biggest challenge is to deploy devices to record real data about the traffic flows, which is subject to time and resource constraints. As a result, we sticked to test the model by using the same dataset, and the aim is to develop a semi-supervised model for future deployment.

Thank you.

Sincerely,

Dazhou Liu